# High-Efficiency Conversion of Bread Residues to Ethanol and Edible Biomass Using Filamentous Fungi at High Solids Loading: A Biorefinery Approach

**Joanna Kawa-Rygielska** [1,*], **Witold Pietrzak** [1] **and Patrik R. Lennartsson** [2]

[1] Department of Fermentation and Cereals Technology, Wrocław University of Environmental and Life Sciences, 51-630 Wrocław, Poland; witold.pietrzak@upwr.edu.pl
[2] Swedish Centre for Resource Recovery, University of Borås, 50190 Borås, Sweden; patrik.lennartsson@hb.se
[*] Correspondence: joanna.kawa-rygielska@upwr.edu.pl

**Abstract:** Bread residues represent a significant fraction of retail food wastes, becoming a severe environmental challenge and an economic loss for the food sector. They are, however, an attractive resource for bioconversion into value-added products. In this study, the edible filamentous fungi *Neurospora intermedia* and *Aspergillus oryzae* were employed for the production of bioethanol and high-protein biomass by cultivation on enzymatically liquefied bread-waste medium at 150 g/L solids. The fermentation of hydrolysate by *N. intermedia* resulted in the ethanol titer of 32.2 g/L and biomass yield of 19.2 g/L with ca. 45% protein. However, the fermentation ended with a considerable amount of residual fermentable sugars; therefore, the liquid medium after the first fermentation was distilled and fermented again by two fungal strains (*N. intermedia* and *A. oryzae*). The fermentations resulted in the production of additional ethanol and biomass. *A. oryzae* showed better performance in the production of biomass, while the other strain yielded more ethanol. The final products' yield ranged 0.29–0.32 g EtOH/g and 0.20–0.22 g biomass/g bread waste depending on the strain used in the second fermentation. The study shows that valorization of bread residuals by fungi is a promising option for the production of biofuels and foodstuff within the circular bioeconomy approach.

**Keywords:** bread residuals; ethanol production; edible biomass; filamentous fungi; *Aspergillus oryzae*; *Neurospora intermedia*; biorefinery



## 1. Introduction

Utilization of residuals, by-products and waste generated during the life cycle of industrial processing of various resources for sustainable production of useful products lies in the basis of the concept of the circular (bio)economy. This is currently considered as a future trend to resolve environmental challenges caused by improper handling of wastes and by-products. One of the core elements of the circular (bio)economy is a biorefinery, defined as an industrial plant converting various organic residuals (biomass) for production of value-added products such as energy, fertilizers, food, chemicals, materials and others [1]. One of the key aspects of a biorefinery is that it utilizes biological agents such as microorganisms and/or enzymes in its technological processes, which catalyzes the process at milder conditions (temperature, pressure) than chemical catalysts or purely physical operations. These biological processes are considered less environmentally challenging than chemical/physical ones through the reduction in the energy and additional processing aids needed for the processing. This results in diminishing the emissions of greenhouse gases and use of toxic chemicals or fossil-based material.

Filamentous fungi (FF) might be considered as good organisms to use as a biorefinery biocatalyst due to their versatility, robustness and ability to degrade organic residue matrixes to simple compounds via their hydrolytic enzyme potential. So far FF were employed at industrial scale for bioproduction of several important products such as enzymes, organic

acids and antibiotics. Moreover, intensive research in this field proved that FF may also be used for production of other valuable products such as biofuels (ethanol and lipids for biodiesel) [2], food/feed grade biomass with high protein content [3], cell wall-based materials [4] and many others [5] and as the agent for biodegradation of soil/water pollutants [6]. Due to the very high market demands, ethanol and edible biomass are probably the most sought products that may be obtained after edible strains of FF fermentation, and they could be produced simultaneously. Ethanol is primarily used as a gasoline additive (a biofuel) but could also be used as a sole vehicle fuel, and it has many other uses as an extraction solvent, synthesis substrate or hand disinfectant, to list some examples. The interest in alternatives to animal protein sources in human diets is currently increasing due to the growing demand for protein. This demand is caused by the ever-growing human population, increased interest in vegetarian/vegan foodstuffs and environmental concerns regarding animal breeding [7]. Therefore, strains of FF that are used in the production of traditional far-east Asia fermented products such as koji (*Aspergillus oryzae*), tempeh (*Rhizopus* sp., *Mucor* sp.) or oncom (*Neurospora intermedia*) could be employed as biocatalysts for simultaneous production of ethanol and food-grade biomass from various organic residuals.

The organic residuals of special interest for biorefining processes are various food wastes (both food processing and retail). According to the recent report, estimated food retail, food service and household wastage ranged around 931 million tons in 2019, where respective waste generation branches are responsible for 12.7%, 26.2% and 61.1% of the whole food waste amount [8]. Bakery products residuals (bread, rolls, pastries) represent a significant fraction of the total food waste generation ranging approx. 12% of the total household food leftovers and 18% of the food retail residuals [9]. This suggests that bakery residues are an abundant and renewable resource for FF biorefinery purposes, and bakery retail wastes can be collected as a relatively homogenous mass composed solely of unsold bread [10]. From a technological viewpoint, bread residuals are also an interesting feedstock. Their principal component is starch, which is a carbohydrate that is relatively easy to degrade to produce fermentable sugars (i.e., glucose). Moreover, many of the fungal strains secrete extracellular amylases; thus, the use of external enzymatic preparations might be vastly limited or not needed at all, reducing the total costs of the processing. Additionally, the drying and subsequent grinding of bread residuals requires far less energy use in comparison to grains [11], which may further contribute to the low processing cost of unsold bakery products.

Previous reports show that bread residuals could be efficiently converted to various bioproducts. Many efforts have been made to show the potential of producing bioethanol from bread waste using yeast, *Saccharomyces cerevisiae*, which, depending on the processing methods applied, can led to high fermentation yields [12,13]. Other products obtained via fermentation processes carried out by various microorganisms include lactic acid [14], succinic acid [15], hydrolytic enzymes (amylases and proteases) [16] and biohydrogen [17]. These studies prove that bakery leftovers are in fact a viable resource for valorization of bread waste within a biorefinery scenario.

This study was aimed at the assessment of the efficiency of bread residue conversion to ethanol together with high-protein biomass by submerged cultivation of edible filamentous fungi strains. The cultivations were performed using high-solids medium (150 g solids per liter) with the feedstock mildly hydrolyzed by enzymatic liquefaction.

## 2. Materials and Methods

### 2.1. Raw Material

Unsold leftovers (after shelf-life) of wheat-rye bread were obtained from a local bakery store. The material showed no visible signs of microbial (i.e., mold) infection. After transportation to the laboratory, the bread loaves were manually cut into cubes of approx. 1–3 cm size and dried overnight in a forced air oven at 50 °C. Then the bread cubes were ground in a Brabender Rotary Mill (Brabender, Duisburg, Germany) with a 1.5 mm internal

mesh sieve. The total dry solids content in the bread (as measured using WPS 50P weighing dryer (Radwag, Radom, Poland)) was $95.8 \pm 0.24\%$ $w/w$, while total starch content (as determined by Evers polarimetric method [18]) was $68.91 \pm 0.48\%$ $w/w$ dry solids basis. The milled feedstock was stored in airtight containers until use.

## 2.2. Microorganisms and Enzyme

A total of four strains of filamentous fungi were used in the study. The two Zygomycetes strains included *Rhizopus oryzae* CCUG 28,958 and *Mucor indicus* CCUG 2242 (Culture Collection University of Gothenburg, Gothenburg, Sweden). Two Ascomycetes strains included *Neurospora intermedia* CBS 131.92 and *Aspergillus oryzae* CBS 819.72 (Centraalbureau voor Schimmelcultures, Utrecht, The Netherlands). All strains were maintained on Petri dishes with potato dextrose agar (PDA; BTL, Łódź, Poland) medium (20 g/L glucose, 4 g/L potato extract, 15 g/L agar) at 4 °C until use. Fresh PDA plates were prepared by incubation of spore suspension at 30 °C for 3–5 days. Fungal spore inoculums were prepared by flooding fresh plates with 20 mL sterile distilled water and releasing the spores with an L-shape glass sterile spreader. The spore count in the spore suspensions were determined microscopically using Bürker chambers. Commercial α-amylase preparation (Liquozyme SC DS) with declared activity of 240 kilo novo units per g was donated by Novozymes (Bagsværd, Denmark). According to the enzyme manufacturer's recommendations, the dosage of the preparation should range 0.013–0.025% $w/w$ for corn grain processing. For the purpose of this study, an average dosage of 0.019% $w/w$ was assumed.

## 2.3. Cultivation Medium Preparation

The bread waste hydrolysate was prepared in an automatic mashing apparatus (LB12+, Lochner Labor + Technik, GmbH, Berching, Germany). An amount of 67.5 g of milled bread (as is basis) was weighted into pre-weighted 500 mL stainless steel mashing beakers and filled with distilled water to 450 mL volume, resulting in a total dry solids concentration of 150 g/L. Then, the beakers were placed in a mashing apparatus water bath preheated to 50 °C and connected to the agitators. Continuous agitation of 150 rpm was maintained throughout the liquefaction process. The bread slurries were heated to 60 °C, and the liquefaction enzyme (Liquozyme SC DS) was added. Approximately one-half of the recommended enzyme dosage was applied in this study. To this end, the enzyme preparation was diluted by weighing 0.5 g of enzyme in a 10 mL volumetric flask and making the volume with distilled water. The diluted enzyme was added to preheated bread slurries at 20 μL volume, resulting in an applied enzyme dose of 0.001% $w/w$. After the enzyme addition, the water bath was heated up to 85 °C at 2 °C/min, and that temperature was maintained for 60 min followed by cooling down to 20 °C. Water loss due to evaporation was replenished with distilled water. A total of 8 beakers were made in this manner simultaneously, and the bread hydrolysates were combined in a 5 L plastic beaker and mixed together followed by pH adjustment to 5.5 using 100 g/L $H_2SO_4$ (Chempur, Piekary Śląskie, Poland) solution. The medium was stored in screw-capped plastic containers at $-20$ °C until used.

## 2.4. Fungal Cultivations

### 2.4.1. Fungal Strain Selection

In the first step of the experiment all four fungal strains were tested for their ethanol and biomass production performances in the bread waste medium. To this end, 100 mL bread hydrolysate aliquots were loaded into 300 mL wide neck conical flasks, plugged with cotton plugs and sterilized (20 min, 121 °C). Upon cooling, the media were inoculated with 1 mL of fungal spore suspensions. Spore counts in the inocula were as follows: $4.15 \pm 0.48 \times 10^6$ for *N. intermedia*, $5.43 \pm 0.67 \times 10^6$ for *M. indicus*, $9.39 \pm 1.12 \times 10^6$ for *R. oryzae* and $5.01 \pm 0.64 \times 10^6$ for *A. oryzae*. Cultivations were performed in duplicate in a water bath shaker (Type 357+, Elpin, Poland) at 35 °C for 72 h. At the end of the cultivations, the biomass yield and the concentrations of sugars and ethanol were determined (see Section 2.4.3) on that basis, two strains were selected for further experiments.

### 2.4.2. Two-Stage Fermentation

The strain that produced the highest ethanol titer was chosen for the primary fermentation in order to maximize ethanol efficiency. To this end, the fermentations were repeated under the same time and temperature conditions in the same manner as described above. Two flasks were prepared for each 24 h time interval at which the biomass, sugars and ethanol were determined. At the sampling point, whole cultures were filtered through 85 μm nylon filtration cloth. Additionally, 6500 mL conical flasks containing 250 mL of bread hydrolysate were inoculated with 2.5 mL of fungal spore suspension. After 72 h of cultivation, the liquid fraction after biomass removal was subjected to distillation as described previously [19]. The stillage remaining in the distillation flasks was cooled down and stored at −20 °C until used. The second fermentation step was conducted using the stillage obtained from the distillation without additional treatment. The two strains used for the second stage were ones that yielded the highest ethanol and biomass amounts. To this end, after thawing of the material, it was poured into 300 mL conical flasks in 100 mL aliquots and sterilized. The flasks were inoculated with 1 mL of fungal spore suspensions, and the fermentation was conducted in the same manner as the first step.

### 2.4.3. Analytical Methods

The biomass content in the fermentation samples was determined gravimetrically. The solid residue obtained after filtration through 85 μm nylon filtration cloth was extensively washed with distilled water and then dried in a forced air oven at 70 °C for at least 24 h. The biomass content was expressed in g/L. Undissolved solids present in the fermentation media were determined in the same way as the biomass content. The total amount of dissolved solids in the fermentation samples was determined by specific gravity determination using the Densito 30PX oscillating U-tube density meter (Mettler-Toledo, Greifensee, Switzerland) and converted to g/L using appropriate tables. The protein content in the biomass was determined by the biuret method using bovine serum albumin (Chempur) as is standard [20]. The liquid samples of the bread hydrolysate as well as the fermentation samples were analyzed for sugars (dextrin (DP4+), maltotriose, maltose and glucose), glycerol, acetic acid, lactic acid and ethanol using high-performance liquid chromatography (HPLC). For this purpose, the samples were centrifuged at 14,000 rpm (11,203× $g$) for 10 min at 4 °C with the MPW 351R centrifuge (MPW Med Instruments, Warsaw, Poland) and filtered using a 0.22 μm nylon syringe filter in the HPLC vials. The samples were analyzed using the Shimadzu Prominence HPLC system (Shimadzu Corp., Kyoto, Japan) equipped with Rezex ROA-Organic Acid H+ column (300 × 7.6 mm) (Phenomenex, Torrance, CA, USA). The separation parameters were eluent used—0.005 M $H_2SO_4$ (Chempur) solution, eluent flow rate—0.6 mL/min and column temperature—60 °C. The compounds were detected using the RID-20A refractive index detector (Shimadzu) maintained at 50 °C. The chromatograms were integrated using CHROMAX 10.0 software (Pol-Lab, Warsaw, Poland) using the external standard method based on a 5-point calibration curve. The product (ethanol, biomass) yields from the substrates were calculated according to the following equation:

$$Y = P/S, \tag{1}$$

where

Y—product (ethanol or biomass) yield ($g_{product}/g_{substrate}$);
P—product concentration in the medium (g/L);
S—substrate concentration in the medium (g/L).

For the first cultivation stage, the S value was the initial concentration of bread residue in the medium, while for the second fermentation stage, the sum of all soluble carbohydrates determined via HPLC was used.

All experiments and analyses were determined in duplicate. The results presented are average values with the error bars or intervals representing one standard deviation ($n = 2$).

## 3. Results

### 3.1. Fermentation Medium Characteristics

The bread hydrolysate properties are presented in Table 1. The dissolved sugars represented almost 93% of the total solubles in the medium. The remaining part is probably composed of soluble proteins and minerals. The most abundant carbohydrate in the bread hydrolysate were dextrins (73.3% of total). The sugars assimilable by typical *S. cerevisiae* (glucose and maltose) represented only 15.5% of the available carbohydrates, which suggests that the use of saccharification enzyme would be necessary to perform an efficient fermentation in this case. Small amounts of fermentation products, i.e., lactic acid and glycerol, were found in the medium. This was a result of the technology of the baking process performed in Poland, which most commonly involves the use of sourdough, a flour slurry fermented by yeast and lactic acid bacteria used to acidify the bread dough.

**Table 1.** Physicochemical features of bread hydrolysate used in this study.

| Component | Concentration (g/L) |
|---|---|
| Dextrin (DP4+) | $95.36 \pm 6.28$ |
| Maltotriose | $14.57 \pm 0.83$ |
| Maltose | $14.63 \pm 0.43$ |
| Glucose | $5.52 \pm 0.03$ |
| Total dissolved carbohydrates | $130.08 \pm 7.52$ |
| Lactic acid | $0.19 \pm 0.06$ |
| Glycerol | $0.12 \pm 0.07$ |
| Dissolved solids | $140.13 \pm 1.13$ |
| Undissolved solids | $17.68 \pm 0.66$ |

Results are average values $\pm$ one standard deviation ($n = 2$).

### 3.2. Fungal Strain Selection for Efficient Ethanol and Biomass Production

The final effects of 72 h cultivation of four studied fungal strains are presented in Table 2. It was observed that *N. intermedia* was the best ethanol producer amongst the studied strains; however, it produced the lowest amount of biomass. The final ethanol titer for this strain reached 4.16% *v/v*. *A. oryzae* cultivation resulted in the highest biomass yield, while producing the least ethanol of all the fungi tested. It was observed that all of the fungal strains resulted in considerable amounts of unfermented sugars ranging 40.48–82.10 g/L in total, which resulted in the sugar consumption ranging approx. 37–69% from the initial medium. *N. intermedia*, *A. oryzae* and *M. indicus* were able to hydrolyze the dextrin from the bread hydrolysate to glucose, which was the most abundant sugar remaining after cultivations. In the case of *R. oryzae*, dextrins were found as the most abundant carbohydrate in the medium followed by maltose. This suggests that the saccharifying enzyme produced by this strain is β-amylase, while for other strains it is glucoamylase. Additionally, *R. oryzae* was the only strain able to produce lactic acid during the fermentation; other fungi utilized the lactate present in the medium from the beginning. *M. indicus* was found to produce the highest glycerol amount, producing nearly twice as much as the rest of the fungi. The biomass produced by all of the fungal strains had a very high protein content at approx. 46%, which suggests that it could be used as an alternative protein source for human consumption. It should be noted that the biomass harvested after each cultivation inevitably contained residual particulate matter present in the hydrolysate from the beginning by entanglement by the fungal mycelium. Additionally, some parts of the particulates were most probably partially hydrolyzed by the enzymes produced by fungi. Therefore, it is very difficult to predict the exact fungal biomass yield from such media; however, from the practical point of view, since the raw material used in this study was food-grade, any remaining particles that were entangled by the fungal biomass would, rather, not be accounted as food contaminants; thus, it would be consumer safe.

**Table 2.** Comparison of the fermentation effects by four edible fungi strains after 72 h cultivation on bread hydrolysate media.

| Concentration (g/L) | *N. intermedia* | *A. oryzae* | *R. oryzae* | *M. indicus* |
|---|---|---|---|---|
| Dextrin | 13.88 ± 1.92 | 10.12 ± 0.84 | 42.65 ± 0.13 | 17.54 ± 0.22 |
| Maltoriose | 5.12 ± 0.11 | 2.66 ± 0.17 | 4.40 ± 0.58 | 6.00 ± 0.10 |
| Maltose | 2.76 ± 0.64 | 3.48 ± 0.09 | 33.58 ± 0.17 | n.d. |
| Glucose | 28.67 ± 2.38 | 24.22 ± 1.85 | 1.48 ± 0.08 | 17.13 ± 2.27 |
| Lactic acid | n.d. | n.d. | 1.93 ± 0.16 | n.d. |
| Glycerol | 1.58 ± 0.01 | 1.22 ± 0.03 | 1.68 ± 0.05 | 2.69 ± 0.01 |
| Ethanol | 32.90 ± 0.74 | 18.55 ± 0.26 | 21.27 ± 0.65 | 29.42 ± 0.42 |
| Biomass | 18.86 ± 0.01 | 25.21 ± 0.41 | 19.50 ± 0.38 | 20.12 ± 0.27 |
| Protein in biomass (g/g) | 0.46 ± 0.02 | 0.47 ± 0.03 | 0.45 ± 0.01 | 0.45 ± 0.02 |
| Ethanol yield (g/g) [1] | 0.22 ± 0.00 | 0.12 ± 0.00 | 0.14 ± 0.00 | 0.20 ± 0.01 |
| Biomass yield (g/g) [1] | 0.12 ± 0.00 | 0.17 ± 0.01 | 0.13 ± 0.01 | 0.13 ± 0.01 |

[1] As g ethanol/biomass per g of initial hydrolysate solids. n.d.—not detected. Results are average values ± one standard deviation (*n* = 2).

Due the fact that none of the studied fungal strains was able to utilize most of the dissolved sugars, it was decided to conduct a two-stage fermentation in which, in the second stage, the liquid residue remaining after the distillation (stillage) of the fermented bread hydrolysate would be subjected to the second fungal cultivation. The screening experiment proved that *N. intermedia* is the highest ethanol producer, so it was chosen for the first fermentation stage and studied in detail. For the second fermentation, two strains were chosen: *N. intermedia* to maximize ethanol yields and *A. oryzae* to improve edible biomass production.

*3.3. Two-Stage Fermentation of Bread Hydrolysate*

Fermentations with *N. intermedia* were repeated and analyzed for efficiency every 24 h. The utilization of dextrin present in the medium started from the beginning of the cultivation, resulting in an almost linear slope of its concentration profile (Figure 1a). Within the first 24 h of fermentation, the concentration of all other analyzed sugars increased by approx. 5 g/L. After that, maltotriose was depleted from the medium, while the concentration of glucose and maltose continued to increase, reaching similar values (approx. 27 g/L) after 48 h. Throughout the final day of the fermentation process, maltose was utilized by the fungus to a final concentration of 3.3 ± 1.6 g/L. At the same time, glucose concentration continued to increase to a final level of 29.6 ± 2.6 g/L, resulting in a nearly six-fold increase in comparison to its initial content. In general, the final concentrations of carbohydrates were similar to those obtained for *N. intermedia* in the screening experiment. It was found that the production of ethanol by the fungus was slow during the first day of fermentation (Figure 1b). Its concentration reached 3.4 ± 0.2 g/L after 24 h with volumetric productivity of 0.14 g/L × h. The corresponding values obtained after 48 and 72 h reached 17.6 ± 1.0 g/L, 0.59 g/L × h and 32.2 ± 2.1 g/L and 0.60 g/L × h, respectively. It was observed that the highest biomass content (26.5 ± 2.2 g/L) was achieved after the first day of fermentation, after which it gradually decreased to a final level of 19.2 ± 0.4 g/L. This observation proves that the fungus was able to hydrolyze the undissolved particles present in the medium. The protein content in the resulting biomass was similar to that determined in the screening experiment, i.e., 0.45 ± 0.01 g/g (data not shown).

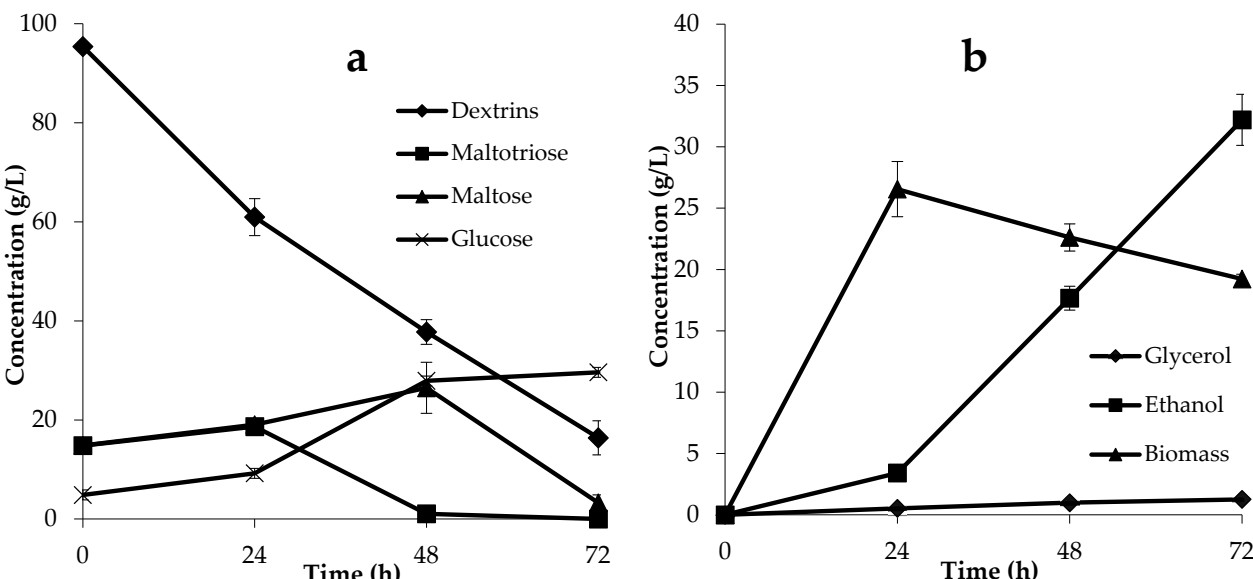

**Figure 1.** Concentration profiles of (**a**) carbohydrate utilization and (**b**) production of ethanol, glycerol and biomass during *N. intermedia* cultivation on bread waste hydrolysate. Results are average values ± one standard deviation (*n* = 2).

The composition of the medium for the second fermentation stage is shown in Table 3. The stillage did not contain a significant amount of undissolved solids, which might have affected the biomass estimation.

**Table 3.** Concentration of sugars and fermentation products in the stillage obtained after fermentation and distillation of bread hydrolysate.

| Component | Concentration (g/L) |
|---|---|
| Dextrin (DP4+) | 19.60 ± 0.05 |
| Maltotriose | 4.80 ± 0.00 |
| Maltose | 4.93 ± 0.01 |
| Glucose | 27.78 ± 0.06 |
| Total dissolved carbohydrates | 57.11 ± 0.10 |
| Lactic acid | n.d |
| Glycerol | 1.41 ± 0.00 |

n.d.—not detected. Results are average values ± one standard deviation (*n* = 2).

The comparison of the stillage fermentation effects by *N. intermedia* and *A. oryzae* is shown on Figure 2. It was observed that both fungi utilized dextrin at a similar rate to a final concertation of 14.2 ± 0.7–14.8 ± 0.1 g/L, with an average consumption of 26%. No significant differences were observed for the profile of maltotriose utilization for both studied strains. *N. intermedia* showed higher efficiency of maltose and glucose utilization from the medium than *A. oryzae*. For both strains, a small (0.2–0.6 g/L) increase in maltose concentration was observed after the first day of fermentation. After that, *N. intermedia* showed a significantly higher maltose utilization rate, resulting in its final concentration of approx. 0.8 ± 0.0 g/L (84% utilized). At the same time, *A. oryzae* utilized maltose at a much slower rate, to a final concentration of 3.2 ± 0.2 g/L with only 34.7% sugar utilized. It was observed that *N. intermedia* intensively utilized available glucose until the second day of cultivation, after which it significantly declined, resulting in a final utilization level of about 96%. *A. oryzae* showed a much slower rate of glucose consumption throughout the process; however, it resulted in a lower final concentration of glucose (0.5 ± 0.0 g/L, 98% utilized). Despite the differences in the particular sugars' utilization rates, the final total content of all analyzed saccharides was similar for both tested strains, ranging 20.2 g/L with 65.6% utilized. It was observed that both studied fungi showed a very

similar profile of glycerol concentration throughout the cultivation processes. As observed during the screening experiment, *A. oryzae* proved to be more efficient at producing biomass, while *N. intermedia* produced more ethanol during the cultivation on bread hydrolysate stillage. The final ethanol and biomass titers were $14.9 \pm 0.1$ g/L and $10.9 \pm 0.1$ g/L for *N. intermedia*, respectively, while the same values for *A. oryzae* were $11.3 \pm 0.7$ and $14.0 \pm 0.3$ g/L. The yield of respective products of cultivation, calculated over the initial concentration of saccharides in the stillage, were 0.26 g/g ethanol and 0.19 g/g biomass for *N. intermedia* and 0.20 g/g ethanol and 0.24 g/g biomass for *A. oryzae*. Unlike during the fungi screening experiment, the biomass of *N. intermedia* contained more protein ($0.49 \pm 0.00$ g/g) than the biomass of *A. oryzae* ($0.43 \pm 0.01$ g/g).

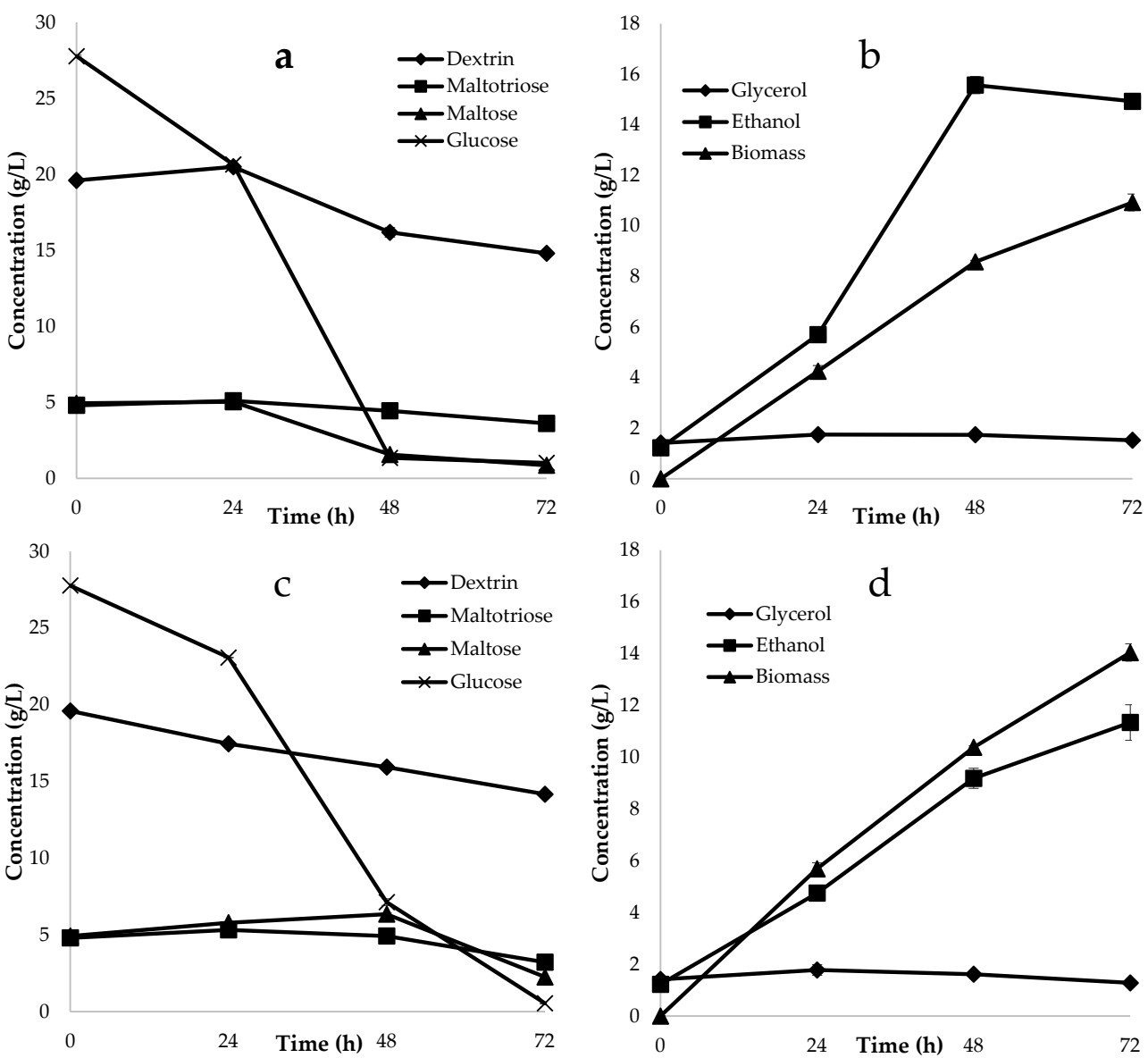

**Figure 2.** Concentration profiles of (**a**,**c**) sugars and fermentation products (**b**,**d**) during cultivation of *N. intermedia* (**a**,**b**) and *A. oryzae* (**c**,**d**) using the stillage obtained after the first fungal fermentation on bread waste hydrolysate. Results are average values $\pm$ one standard deviation ($n = 2$).

### 3.4. Process Mass Balance

On the basis of the results obtained during the cultivation process, the mass balance was calculated for the processing of 1000 g of bread residue used (Figure 3). After summarizing the product streams obtained after both cultivation stages, the ethanol, biomass

and protein yields ranged 315.7, 201.8 and 93.8 g/kg bread residue, respectively, when *N. intermedia* was applied for both cultivation stages. The respective values in the case of the second cultivation stage with *A. oryzae* were 291.7, 222.7 and 98.3 g/kg bread. It should be pointed out that the first cultivation stage provided the majority of the product masses obtained. Nevertheless, the application of the second fermentation stage improved the overall ethanol yield by 26.0% and 31.7% for *A. oryzae* and *N. intermedia*, respectively. In the case of protein production, the second fermentation stage improved the process mass balance by 38.3% and 41.1% for *N. intermedia* and *A. oryzae*, respectively. Therefore, the second cultivation stage was crucial for the overall efficiency of the entire process.

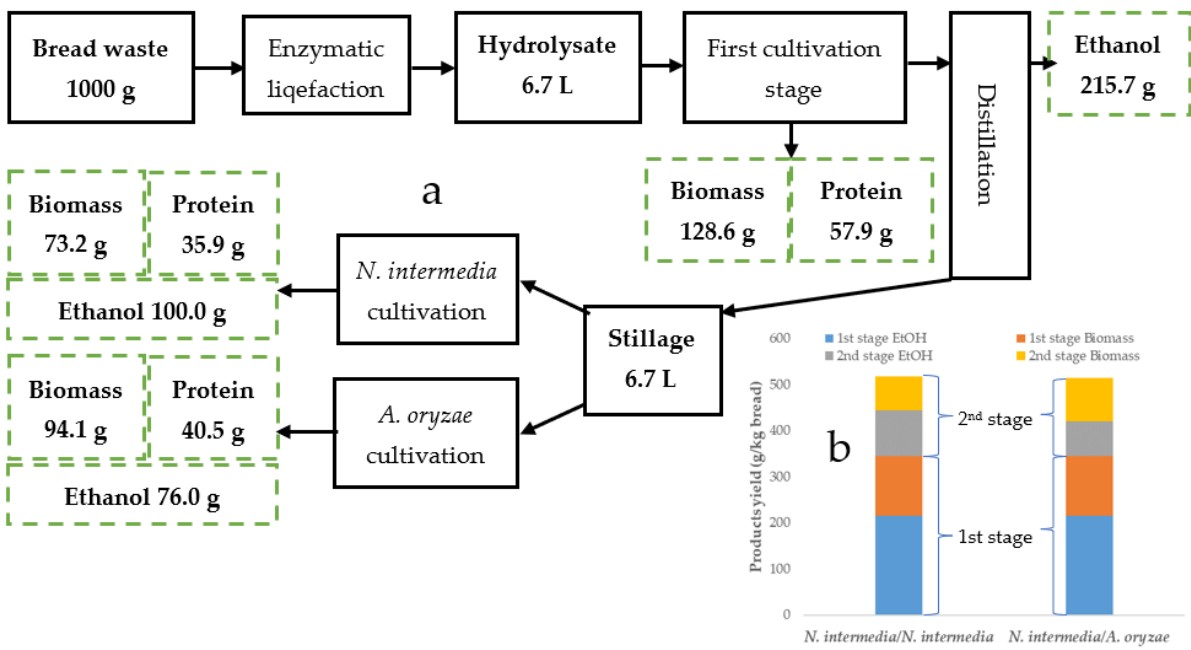

**Figure 3.** Mass balance for the process of bread waste valorization using edible filamentous fungi (**a**). The streams of the products obtained are contained in the dashed green line blocks. The bar graph (**b**) shows the comparison of fungal biomass and ethanol production in both fermentation stages depending on the fungal strain used.

## 4. Discussion

In this paper, bread residuals were converted into ethanol and valuable high-protein biomass for food or feed uses. According to the mass balance, the final ethanol yield ranged approx. 29.2–31.6% by weight. In previously published papers dealing with conversion of bread waste to ethanol, the reported product efficiencies were higher depending on the raw material pretreatment applied. Pietrzak and Kawa-Rygielska [21] studied conversion of bread to ethanol by *S. cerevisiae* with the application of granular starch hydrolyzing enzyme at the same substrate concentration in the medium as applied here (150 g/L) and reported ethanol yields between 35.4–38.6% *w/w* with an average ethanol titer of 55 g/L. Studies on the simultaneous saccharification and fermentation of bread hydrolysates at higher solids (>30% *w/v*) loading by the yeast proved to be even more effective, yielding nearly 40% *w/w* bread to ethanol conversion efficiencies [13]. Recently, Narisetty et al. [22] studied the comparison of acid and enzymatic hydrolysis of bread residuals on ethanol production by *S. cerevisiae* via separate hydrolysis and fermentation. They showed that enzymatic hydrolysis results in higher ethanol yield than acid hydrolysis (176 g/Kg vs. 240 g/Kg). The use of acid as hydrolysis catalyst seemingly might be a good option for releasing glucose from starchy residuals due to short processing times needed. However, as shown by Narisetty, high processing temperature (121 °C) and presence of acid catalysts results in the production of high levels of toxic hydroxymetylfurfural (up to 2.5 g/L) and furfural (up to 0.7 g/L), which could inhibit the fermenting microorganism as well as

diminish the use of the fermentation by-products for food or feed use. Acid hydrolysis might, however, be an effective method for mycotoxin (namely aflatoxins) removal from contaminated bread residuals, which further improves ethanol production in comparison to enzymatic hydrolysis [23]. Nevertheless, it is well-known that the application of *S. cerevisiae* in the production of ethanol from starchy materials requires the use of saccharification enzymes for the hydrolysis of dextrins formed during the liquefaction stage to glucose. As analyzed by Kwiatkowski et al. [24], the cost of enzymes used in the production of bioethanol from corn represent up to 4.9% of the total operation costs of the ethanol plant. In this paper, it was shown that the use of *N. intermedia* as the sole microorganism in the production of ethanol from bread residuals may be efficiently carried out without the use of any saccharification enzyme and only one-half of the recommended liquefaction enzyme dose. This would positively impact the economic indicators of the process described in the present study. It could also be speculated that the application of $\alpha$-amylase might be omitted, and the unpretreated bread slurry would be fermented by the fungi, which are capable of degrading starch, as shown by the carbohydrate utilization profiles in this study. This, however, might result in the generation of medium with exceptionally high viscosity and undissolved particles, the pumping and handling of which would be difficult. The use of bacterial $\alpha$-amylase, besides effective viscosity reduction in starchy materials, also provides the advantage of very high optimal temperature of the activity of this enzyme, which often exceeds 80 °C [25]. This means that the enzymatic liquefaction would serve as a process where viscosity reduction, provision of small amounts of simple sugars to initiate fungal growth and microbial stabilization would take place, thus reducing the amount of process operations and equipment needed.

The ethanol titer produced by *N. intermedia* during the first fermentation stage (32.2 g/L) is probably one the highest reported by this species. In the previous studies with starchy residuals, the reported ethanol concentrations ranged approx. 5 g/L (wheat-based thin stillage) [26], approx. 8 g/L (liquid obtained after hydrothermal pretreatment of brewer's spent grains) [27] and approx. 18 g/L (acid-pretreated wheat bran) [28]. This proves that bread waste is a more efficient residue that does not require harsh conditioned pretreatment. It could also be speculated that longer first stage fermentation would result in even higher ethanol production because no production rate decline was observed on the ethanol concentration curve (Figure 1b). Additionally, previous attempts at producing ethanol on bread waste by filamentous fungi were less efficient than the one reported here. Svensson et al. [29] used *R. oryzae* to produce ethanol and nutritional biomass from bread residue without the application of any material pretreatment except sterilization. They obtained a maximal ethanol concentration of roughly 17 g/L at 15% bread concentration in the medium. In this study, *R. oryzae* was able to produce 21.3 g/L ethanol at the same bread content, which shows that the enzymatic liquefaction of the substrate improves the fermentation efficiency. Moreover, bread residues could also be used for production of foodstuffs using FF growing in solid state fermentation. For example, Gmoser et al. [30] analyzed bread waste fermentation by *N. intermedia* under solid state conditions, which resulted in obtaining a 'fungal burger' with acceptable sensory features and enhanced nutritional properties, e.g., three-fold protein content increase, nine-fold dietary fiber content increase and substantial production of vitamins (D2, E and C). Melikoglu et al. [31] used bread waste to produce protease and glucoamylase by A. awamori also under solid state conditions. The fermented product exhibited 114 U/g glucoamylase and 83.2 U/g protease activities, which shows the potential of bread residues for fungal enzyme production as well.

It is known that the by-products play a crucial role in the economical balance of ethanol production [22]. The major distillery by-product is the stillage, which, after drying, is sold as a high-protein feed additive known as dried distiller's grains with solubles (DDGS). However, little is known about the protein content of bread waste DDGS. Wheat (major substrate for bread baking) based DDGS contains as much as 38% protein [32]. In this study, fungal biomass with the protein content exceeding 45% was obtained as a major by-product. Therefore, edible filamentous fungi might be a more efficient option for mitigating protein

deficiency than plant-based protein isolates. Moreover, fungal fermentations result in the production of other valuable products (such as ethanol, which was presented in this study), while isolation of plant protein generates by-products in the form of non-protein fractions, which could also be utilized by fungi [33].

## 5. Conclusions

In this study, it was shown that the use of edible filamentous fungi can be applied for efficient conversion of bread residues into value-added products such as ethanol and high-protein biomass for food or feed applications. The cultivation of fungi resulted in the high amounts of residual mono- and disaccharides; therefore, a two-stage fermentation process was applied to maximize the yield of the products. The best ethanol-producing strain proved to be *Neurospora intermedia*, while *Aspergillus oryzae* was able to produce the highest amount of biomass. The final bread-to-ethanol conversion efficiencies were just slightly lower than the ones reported for yeast fermentations, with the advantages of producing protein-rich biomass for human/animal consumption and less processing steps needed for the conversion process.

**Author Contributions:** Conceptualization, J.K.-R., W.P. and P.R.L.; methodology, W.P.; validation, W.P.; formal analysis, W.P.; investigation, W.P.; resources, W.P. and P.R.L.; data curation, W.P.; writing—original draft preparation, W.P. and P.R.L.; writing—review and editing, J.K.-R., W.P. and P.R.L.; supervision, J.K.-R. All authors have read and agreed to the published version of the manuscript.

**Funding:** This work was supported by the Wrocław University of Environmental and Life Sciences (Poland) for the Ph.D. research program "Innowacyjny Naukowiec", No. N060/0011/21.

**Institutional Review Board Statement:** Not applicable.

**Informed Consent Statement:** Not applicable.

**Data Availability Statement:** Not applicable.

**Conflicts of Interest:** The authors declare no conflict of interest.

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
