# Peer review of "High-Efficiency Conversion of Bread Residues to Ethanol and Edible Biomass Using Filamentous Fungi at High Solids Loading: A Biorefinery Approach"

_applsci, doi:10.3390/app12136405_

Round 1

Reviewer 1 Report

The manuscript describes the production of biofuels (ethanol) and edible protein from bread residues through fermentation with edible fungi. Of course, the idea is of significance, however only of limited novelty. Nonetheless the study is scientifically sound and the data are presented clearly and understandably. One remark in this respect: Sometimes very long sentences make it complicated to understand what the authors wanted to say. Please consider revising and just split some of the long sentences. This should not take much time, but will improve the readability.

M&M:

- origin of chemicals (manufacturer,etc.) is missing.

- providing the mathematical formula which was used for the calculation of EtOH yield per g saccharide or “bread substrate” would improve the quality of the manuscript

General:

-      It seems that the experiments were performed once in duplicate. Are there really no replicate experiments? Are the data obtained from a “one time observation” only?

-      "ca." should be replaced by an appropriate english term, e.g. "approx." throughout the manuscript.

Results:

-      In the description of the figures in the results part, values should be noted with their respective Std.dev., e.g. 3.3 ± 0.3 g/L (as done in line 298).

-      Line 270: it would be better (more consistent) if medium composition was shown as under 3.1 in a table.

-      Figure 3: splitting into subfigures (a,b) could help to improve the presentation to the reader.

Discussion:

The discussion should be improved by comparing with (and citing) more of the availably literature on fermentation of bread residuals (e.g. in terms of yields and microorganisms used). There are many more articles dealing with fermentation of bread by filamentous fungi that deserve to be compared to the data presented in the manuscript.

Reviewer 2 Report

1. Few sentences seemed to be incomplete:

Line 277-278: “N. intermedia shown higher efficiency of maltose and glucose utilization from the medium than A. oryzae.”

Line 279-281: “After 279 that, N. intermedia shown a significantly higher maltose utilization rate resulting in its final 280 concentration of ca. 0.8 g/L (84% utilized).”

2. Would it be appropriate to represent the product yields of biomass and ethanol all together in one bar? In other words, biomass and ethanol could be considered separately, not as accumulative factors.

3. In Figure 3, it is somewhat confused to see the text boxes with solid line and with dotted line. Differentiation with other factors such as box shape or color would be helpful for readers to understand it clearer.  

4. More comparison with other studies would be required. In other words, the number of references in Discussion part seemed insufficient.

5. The authors described that biomass is possibly inaccurate due to the solid matter from bread hydrolysate. Is there any reason for not filtrating?  
